# A Study on Heat Storage and Dissipation Efficiency at Permeable Road Pavements

**DOI:** 10.3390/ma14123431

**Published:** 2021-06-21

**Authors:** Ching-Che Yang, Jun-Han Siao, Wen-Cheng Yeh, Yu-Min Wang

**Affiliations:** 1Department of Civil Engineering, National Pingtung University of Science and Technology, Pingtung 91201, Taiwan; a000550@oa.pthg.gov.tw (C.-C.Y.); p10833002@g4e.npust.edu.tw (J.-H.S.); weyeh@mail.npust.edu.tw (W.-C.Y.); 2General Research Service Center, National Pingtung University of Science and Technology, Pingtung 91201, Taiwan

**Keywords:** porous asphalt concrete, permeable road pavement, temperature distributions, urban heat island effect

## Abstract

The main contributing factor of the urban heat island (UHI) effect is caused by daytime heating. Traditional pavements in cities aggravate the UHI effect due to their heat storage and volumetric heat capacity. In order to alleviate UHI, this study aims to understand the heating and dissipating process of different types of permeable road pavements. The Ke Da Road in Pingtung County of Taiwan has a permeable pavement materials experiment zone with two different section configurations which were named as section I and section II for semi-permeable pavement and fully permeable pavement, respectively. The temperature sensors were installed during construction at the depths of the surface course (0 cm and 5 cm), base course (30 cm and 55 cm) and subgrade (70 cm) to monitor the temperature variations in the permeable road pavements. Hourly temperature and weather station data in January and June 2017 were collected for analysis. Based on these collected data, heat storage and dissipation efficiencies with respect to depth have been modelled by using multi regression for the two studied pavement types. It is found that the fully permeable pavement has higher heat storage and heat dissipation efficiencies than semi-permeable pavement in winter and summer monitoring period. By observing the regressed model, it is found that the slope of the model lines are almost flat after the depth of 30 cm. Thus, from the view point of UHI, one can conclude that the reasonable design depth of permeable road pavement could be 30 cm.

## 1. Introduction

With the development of cities, the population requires more facilities, which leads to more roads and building construction [1]. Urban heat island (UHI) is a commonly observed phenomenon worldwide [2]. The formation of UHI can be mainly ascribed to the increased absorption and trapping of solar radiation in built-up urban fabrics, and to other factors, including population density of built-up areas and vegetation fractions [3]. It is an urban area where temperatures are significantly higher than those in the surrounding areas [2]. Therefore, urban environmental problems will inevitably become more serious and important [4]. Previous studies about the UHI effect showed that daytime heating is the main contributing factor [5,6]. Some studies have also found that the high temperature in the summer caused by UHI [7], that will cause public health problems such as the increase of heat strokes during the day [8], and an increase in the number of tropical nights leads to an increase in the morbidity and mortality of the elderly or people with chronic diseases [9,10].

Pavement as a key contributor to higher surface and air temperatures have been pointed by researchers [6,11,12,13,14]. Many references mentioned that traditional pavement had high thermal conductivity properties tend to absorb large amounts of solar radiation during daytime and to release it to the cooler surrounding atmosphere subsequently at night, further the formation of the UHI phenomenon [10,14,15,16,17,18,19]. Thus, the Environmental Protection Agency (EPA) and many researches have recommend permeable pavements to mitigate UHI [3,19,20,21,22,23], and, all around the world, many research results have been presented and compared temperatures of traditional pavements and various permeable pavements whether their tests in fields or in laboratories as shown next the paragraph.

In Taiwan, the in-situ study by Cheng et al. [18] and Paramitha et al. [24] utilized T-type thermocouples to obtain temperature fluctuations in the diurnal cycle of dense-graded asphalt concrete, permeable asphalt, permeable concrete, and permeable interlocking concrete block pavements in summer. From temperature fluctuations in diurnal cycle results by Cheng et al. [18] pointed out that, due to solar radiation can penetrate deeply into the structural porosity of porous asphalt concrete pavement, lead to the peak temperatures at near-surface of it was higher than that of dense-graded asphalt concrete pavement. In addition, the results by Paramitha et al. [24] found that temperature between pavements different, especially at noon. Permeable pavements had lower temperature than impermeable pavement at noon. In Japan, Tokyo, from Asaeda and Ca’s field experiment results [25] found that the porous concrete pavement had higher maximum temperatures, but lower minimum temperatures relative to the traditional concrete pavement in summer. In a parking lot of Iowa, Kevern et al. [26] also utilized temperature sensors to obtain temperature behaviors inside of traditional portland cement concrete and portland cement pervious concrete. The research results conducted that portland cement pervious concrete was hotter than the traditional portland cement concrete during daytime. However, the portland cement pervious concrete was cooler than traditional concrete pavement during nighttime. In Phoenix, Arizona, Stempihar et al. [23] also conducted temperatures comparison of dense graded asphalt, porous asphalt, and porland cement concrete pavements. It was observed that porous asphalt had higher daytime surface temperatures and lowest nighttime temperatures than other materials. In addition to the field tests, the researchers also obtained similar results in the laboratory.

In the laboratory study by Hassne et al. [19], temperature of asphalt slabs with different air voids contents had been measured by J-type thermocouples. The outcome was found that the asphalt mixture slab with high air content is easier to reach a stable temperature than that the low air content. It also be found that asphalt mixture with high air voids content heating and dissipation rates higher than that with the low air voids content. Similarly, the laboratory-simulated study by Wu et al. [22] also showed that the porous surface of the open-graded specimen had lower temperatures than air temperatures during nighttime. Open-graded mixture also had better dissipation effect and declined faster than the dense-graded specimen, especially under wind convection conditions. According to the reference, the wind convection and evaporation in the environment can help to temperatures of permeable pavements decline rapidly and increase dropping rate [22]. Haselbach et al. [27] showed that after a rainfall, porous concrete cooled more rapidly than conventional concrete, thus indicating that evaporative dissipation was occurring. As anticipated, ability of porous asphalt to exhibit cool temperatures at nighttime by quickly dissipating high daytime temperatures makes it an excellent tool to mitigate the UHI effect.

In addition, a one-dimensional pavement model by Stempihar et al. [23] to measure surface temperatures of porous asphalt, dense-graded asphalt, and portland cement concrete. Results showed that porous asphalt had higher surface temperatures than others in daytime. However, porous asphalt had a lower surface temperature than others at nighttime. Even though temperatures of the porous asphalt pavement were rapidly increasing as air temperatures increase, but the temperature of the porous asphalt pavement was rapidly decreasing as air temperatures decrease. Consistent with results of Cheng et al. [28], porous pavements had lower temperature than those for traditional pavements on very hot days. Thus, the large-scale applications of porous pavements could help mitigate urban heat island impacts.

As mentioned in front, permeable pavements could effective to aid UHI effect mitigation. According to the Stempihar et al. [23] study, it becomes important to evaluate the entire pavement structure and material properties when selecting pavement materials to mitigate urban heat island. In the present study constructed two kinds of porous asphalt road pavements, called that fully permeable and semi-permeable. These pavements differ between their materials under the subgrade. Moreover, the temperature sensors were embedded in two permeable road pavements for monitoring the temperature behaviors. In the past, these the storage and dissipation efficiencies of permeable road pavements have not been given much attention during hotter and cool period and are not easily available in the literature. Thus, with the objective of understanding the heating and dissipating process of permeable road pavements. The heat storage and dissipation efficiencies of pavements were calculated by formulas and using regression models to find out the relationships of pavements depths verse those. To further recommend the best permeable road pavement for mitigating UHI effect.

## 2. Materials and Methods

### 2.1. Study Site

The study site, Ke Da Road, is approximately 10 km in south of Pingtung City, Taiwan. The Ke Da Road is about 5.4 km in length, starting from the Formosa Highway and towards the National Pingtung University of Science and Technology (NPUST) Campus. The experiment zone in the present study is 0.2 km long as shown in Figure 1. Two different kinds of permeable road pavements were constructed by measuring 100 m in length * 8.5 m in width with a surface slope of 2%. The pavements were diverse such that section I is semi-permeable while section II is fully permeable. As Figure 2 presented, two type permeable road pavements had the same material configuration at surface course, base course, and subgrade. The material is Porous Asphalt Concrete (PAC) at the surface course, Permeable Concrete (PC) at the base course, and filter layer at the subgrade, respectively. The material physical property of semi and fully permeable pavements is listed in Table 1. In addition, under subgrade configuration where there is an impermeable cloth in section I while a geotextile in section II. In general, high density polyethylene geomembrane (HDPE) impermeable cloth is used in the slopeland engineering more frequently than used in the road engineering. The HDPE impermeable cloth is used to block the infiltration of water. However, the geotextile in Section II, which was woven with polyolefin fibers, and has good water conduction and filtration. The main purpose of using the HDPE impermeable cloth and the geotextile in permeable road pavements in this study is to compare the difference for heat storage and dissipation capabilities under permeable and non-permeable conditions. These geotechnical material specifications are listed in Table 2 and Table 3.

### 2.2. Temperature Data Collection

In order to observe the temperature behaviors of permeable road pavements, five type K thermocouple sensors (from Campbell Scientific, Inc.^®^, Logan, UT, USA and with measurement temperature range from −55 °C to 125 °C) were embedded into the pavements. As shown in Figure 3, the sensors were embedded within pavement at depths of 0, 5, 30, 55 and 70 cm. The first temperature sensor was installed on the surface for monitoring the surface temperature. The second and fifth temperature sensors were embedded in the middle of the materials for monitoring the temperatures of the porous asphalt concrete and the filter layer, respectively. In addition, the third and fourth temperature sensors were arranged at equal distances depths, 25 cm and 50 cm, are apart from the second sensor to investigate the heat conduction effect of various materials going forward. On the other hand, a long term investigation was carried out from January to December 2017. Temperature data of the different permeable road pavements were automatically recorded and collected by the CR1000 data logger (from Campbell Scientific, Inc.^®^) at hourly intervals. The air temperatures of the environment were logged in NPUST weather station. The analysis covers days with the negligible precipitation and relatively higher air temperatures in 2017, which are extreme weather conditions for UHI impact. Based on many studies, the UHI effect has been investigated during the summer [14,18,21,23,24,25,26,27]. However, researchers [21,23] suggested that cooler climates should be considered. Thus, temperature distributions of permeable road pavements in January and in June are taken into account in this study. Using data with no rainfall in January and June to represent the variation of temperatures along different pavement depths in winter and summer.

### 2.3. Calculation of Heat Storage and Dissipation Efficiency

In the present study, Equation (1) is adopted to evaluate the heat storage efficiency and heat dissipation efficiency. The heat storage efficiency and heat dissipation efficiency were defined as the average gradient of temperature with respect to corresponding time intervals. The rising and dropping limbs of the temperature profile in Figure 4, Figure 5, Figure 6, Figure 7 are used in the Equation (1) to estimate the heat storage efficiency and heat dissipation efficiency, respectively. Equation (1) is used to calculate heat storage efficiency and heat dissipation efficiency for the pavements in this study. Heat storage efficiency is that the highest temperature of the day subtracts the lowest temperature, then the result is divided by time intervals of the rising limb. On the other hand, heat dissipation efficiency is the lowest temperature minus the highest temperature of the day, then the result is divided by time intervals of dropping limbs.
(1)$$\mathrm{Heat}\;\mathrm{storage}\;\mathrm{efficiency}\;\mathrm{and}\;\mathrm{Heat}\;\mathrm{dissipation}\;\mathrm{efficiency}=\frac{T_{max}-T_{min}}{\Delta t},$$
where *T_max_* is the maximum temperature, °C; *Tmin* is the minimum temperature, °C; Δt is the corresponding time intervals, h.

### 2.4. Regression Analysis of Permeable Road Pavement Temperatures with Respect to Depths

In order to understand the heating and dissipating capacities of permeable road pavements, heat storage and dissipation efficiencies obtained from Equation (1) were employed as a basis for modelling, the method is established by regression analysis using the heat storage and dissipation efficiencies for various depths of 22nd to 28th in January and 21st to 25th in June of 2017 at the experiment zone as dependent variables.

## 3. Results and Discussion

### 3.1. Temperature Distribution of Permeable Road Pavements in January

This study agrees with previous studies [17,23,26,28,29] which found that the temperature at each depth changed basically synchronously with the air temperature are shown in Figure 4 and Figure 5. Amplitudes of pavements temperature variations decreased while the depth increased. Results showed that temperatures of sections I and II always warmer than air temperature during the hottest period of the day. Due to heat absorption capacities of permeable road pavements leads to their surface temperatures are higher than air temperature. Both sections subsequently cooled than air temperatures in the early morning. In addition, the results also showed that the mean surface temperature of section II approximately 1 °C was warmer than that section I. Although the mean surface temperature of section II was higher than that of section I during the day, the mean surface temperature of section II approximately 3 °C cooled than that of section I early in the morning. It is found that the dissipation capacity of fully permeable road pavement is better than that of semi-permeable road pavement in winter.

### 3.2. Heat Storage and Dissipation Efficiency in January

As listed, Table 4 is a summary of heat storage and dissipation efficiencies at each depth for permeable road pavements, from 22nd to 28th in January. The results showed that heat storage efficiencies of section I were 5.3, 4.6, 2.4, 2.6, and 2.8 °C/h at depths of 0, 5, 30, 55, and 70 cm, respectively. It should be noted that the heat storage efficiency of the surface was higher than that of other depths in section I. However, the heat storage efficiencies gradually decreased as the depth increases until the depth of 30 cm. Under the continuous pavement heat conduction, the heat storage efficiencies at depths of 55 cm to 70 cm in section I slightly increased. The results also showed that heat dissipation efficiencies at the surface of section I had higher about 2 °C/h than other depths. Below the surface course, heat dissipation efficiencies almost the same in the base course and the subgrade with heat dissipation efficiencies about 1 °C/h.

On the other hand, it was observed that section II had high heat storage efficiency about 5.8 °C/h at surface. However, the heat storage efficiency at the depth of 5 cm was lower about 2 °C/h than the heat storage efficiency at the surface. Although the surface heat storage efficiency of section II was very high, the heat storage efficiency at the depth of 5 cm in section II decreased a lot. Subsequently, the heat storage efficiency gradually becomes the same as the depth increases. In terms of the heat dissipation efficiency of section II, the heat dissipation efficiency was 2.3 °C/h at the surface, which was higher about 1 °C/h than other depths. However, the heat dissipation efficiencies at depths of 5 cm to 70 cm in section II almost no difference.

In comparison with two sections, the heat storage and dissipation efficiencies at depths of 0 cm, 30 cm, 55 cm, and 70 cm of section II were higher than that of section I. Hence, it is found that heat storage and dissipation capacities of fully permeable road pavement performs is better than that of semi-permeable road pavement in winter.

### 3.3. Temperature Distribution of Permeable Road Pavements in June

Using consecutive five days data with no rainfall in June to represent temperature variation along pavement depths in summer. As shown in Figure 6 and Figure 7, the temperature at each depth changed basically synchronously with the air temperature from June the 21st to the 25th. The results showed that temperatures of sections I and II always warmer than air temperature during the hottest period of the day. However, temperatures of two permeable road pavements were similar with air temperatures in the morning. In addition, the mean surface temperature of section II was warmer approximately 5 °C than of section I. Similar findings have also documented by previous research [30]. Meanwhile, the mean surface temperature of section II cooled approximately 1 °C than that of section I in the morning. In summer, fully permeable road pavement had better heat dissipation comparing with semi-permeable road pavement.

### 3.4. Heat Storage and Dissipation Efficiency in June

The summary of heat storage and dissipation efficiencies of two sections from 21st to 25th in June are listed in Table 5. The results showed that the heat storage efficiency of section I was 4.3 °C/h originally decreased to 2.1 °C/h as the depth increased. In addition, the results also showed that the heat dissipation efficiencies at the depth of 0 cm of section I was similar with the heat dissipation efficiencies at the depth of 5 cm in section I. However, below the surface course, the same heat dissipation efficiencies of 0.7 °C/h in section I.

On the other hand, it was observed that section II had high heat storage efficiency of about 5.0 °C/h on surface. Although the surface heat storage efficiency of section II was very high, the heat storage efficiency at the depth of 5 cm in section II decreased a lot by comparing the surface. Then, the heat storage efficiency gradually became the same as the depth increased. It was also observed that the heat dissipation efficiency of section II gradually decreased as the depth increased.

In comparison with two sections, the heat storage efficiency at the depth of 0 cm in section II was higher than section I. However, the heat storage efficiencies at the depth of 5 cm in section II was lower than that in section I. As the depths increased, the heat storage efficiencies of section II almost the same with section I. Thus, more heating store on the surface of fully permeable road pavement than semi-permeable road pavement in summer. However, section II has higher heat dissipation efficiencies comparing with section I. The dissipation capacity of fully permeable road pavement is better than in semi-permeable road pavement in summer.

### 3.5. Modelling Permeable Road Pavement Temperature with Respect to Depth

The relationship of heat storage efficiencies and depths in semi and fully permeable road pavements are illustrated in Figure 8a,b. From the trend, it shows that the heat storage efficiency decreases while the depth increases. It can be clearly seen from the figures that the heat storage efficiencies of the upper course in semi and fully permeable pavements are higher than that in the lower courses for winter and summer. It is also found that the slope of the regressed model line is almost flat after depth equal to 30 cm. In addition, the regressed model line for winter is higher than that for summer. Also, the permeable road pavements storage less heat in summer comparing with in winter. Meanwhile, multi regression models were developed for estimating heat storage efficiencies at different depths of semi and fully permeable road pavements as presented by Equations (2) and (3), respectively. It can be seen that these two models will have good performance because their coefficients of determination ($R^{2}$) are 0.94 and 0.97.

The model predicting heat storage efficiency at specified depth can be presented by the following equation:(2)$$y_{\mathrm{IS}}=-0.77226\ln(x)+0.00984x+0.68D+4.59208,$$
(3)$$y_{\mathrm{IIS}}=-1.0439\ln(x)+0.02974x+0.84D+4.81222,$$
where $y_{\mathrm{IS}}$ is the heat storage efficiency of section I for winter and summer, °C/h; $y_{\mathrm{IIS}}$ is the heat storage efficiency of section II for winter and summer, °C/h; x is the specified pavement depth, cm; D is dummy variable that is 1 for winter and 0 for summer.

On the other hand, the relationship between heat dissipation efficiency and depth of semi and fully permeable road pavements are shown in Figure 9a,b. It is observed that heat dissipation efficiencies of permeable road pavements decrease while the depth increases. Obviously, the heat dissipation efficiencies in the surface course of semi and fully permeable pavements are larger than in the lower courses of semi and fully permeable pavements in winter and summer. It is also found that the slope of the regressed model line is almost flat after depth equal to 30 cm. One can see in these Figures, the regressed model line for winter are slightly higher than that for summer in each permeable road pavement. It is clear that no matter which season, the heat dissipation efficiencies of permeable road pavements are similar. In addition, the heat dissipation efficiency verse depth of fully permeable road pavement performs better than that of semi-permeable road pavement. For example, when the dissipation efficiency is 1 °C/h for semi-permeable road pavement at 25 cm depth, however the same dissipation efficiency rate is found at 30 cm depth in fully permeable road pavement, as shown in Figure 9a,b. By introducing dummy variable for winter and summer seasons, multi-regression models have been developed to form heat dissipation efficiencies at different depths of semi-permeable and fully permeable pavements as shown by Equations (4) and (5), respectively. It is observed that R^2^ are 0.95 and 0.91, indicating that two models will have good performance.

The model predicting heat dissipation efficiency at specified depth can be presented by the following equation:(4)$$y_{\mathrm{ID}}=-0.37861\ln(x)+0.00179x+0.12D+2.16698,$$
(5)$$y_{\mathrm{IID}}=-0.4639\;\ln(x)+0.00763x+0.06D+2.32511,$$
where $y_{\mathrm{ID}}$ is the heat dissipation efficiency of section I for winter and summer, °C/h; $y_{\mathrm{IID}}$ is the heat dissipation efficiency of section II for winter and summer, °C/h; x is the specified pavement depth, cm; D is dummy variable that is 1 for winter and 0 for summer.

## 4. Conclusions

In this paper, two permeable road pavements, including semi and fully permeable road pavements, were monitored for temperatures at the depths of 0, 5, 30, 55, and 70 cm in January and June. Based on the collected temperature data, heat storage and dissipation efficiencies with respect to depth have been studied and modelled by using multi regression for the studied pavements. Through the collected and modelled results, the following conclusions are obtained:From temperature results in the two seasons, the maximum surface temperatures of permeable road pavements are higher than those of air temperatures in daytime. Meanwhile, the minimum surface temperatures of permeable road pavements are lower than that of the air temperature at early morning. It is also found that temperature variations of permeable road pavements have the same trend as air temperatures.According to the calculated heat storage and dissipation efficiencies, it is revealed that the heat storage and dissipation efficiencies of fully permeable road pavement are higher than those of semi-permeable road pavement in both January and June. No matter which season, it is found that heat storage and dissipation capacities with respect to depth in fully permeable road pavement performs better than that in semi-permeable road pavement for UHI.Through the results of modelling, urban heat island effect is easily to occur in summer. Reduction rates in depth of heat storage and the dissipation efficiencies in fully permeable road pavement has better capacities in comparison with those of semi-permeable road pavement in summer. Thus, this further proves that using fully permeable road pavement is better than semi-permeable road pavement in hotter environments to aid urban heat island effect.Multi regression models were developed to relate the heat storage and dissipation efficiencies and depth for semi and fully permeable road pavements in this study. It is found that the slope of the regressed model lines is almost flat after the depth of 30 cm. Thus, from the view point of UHI, one can conclude that a reasonable design depth of permeable road pavement could be 30 cm.

## Figures and Tables

**Figure 1 materials-14-03431-f001:**
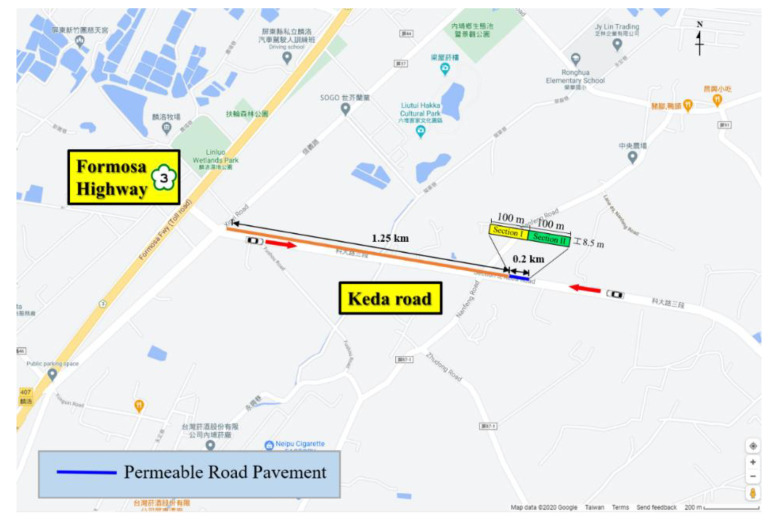
Location of the study area.

**Figure 2 materials-14-03431-f002:**
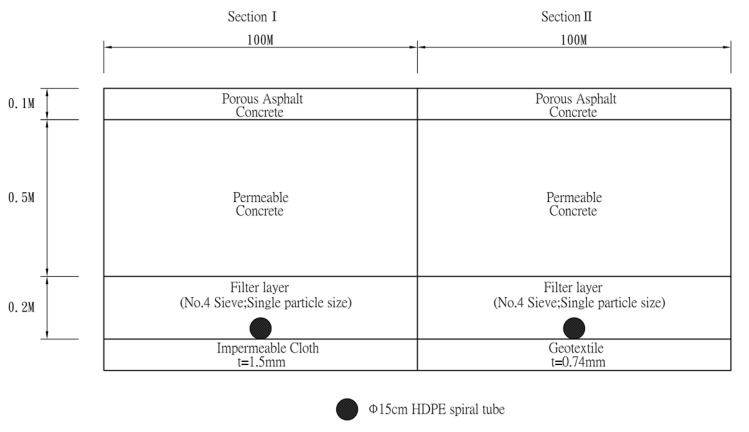
The pavement designs of this study.

**Figure 3 materials-14-03431-f003:**
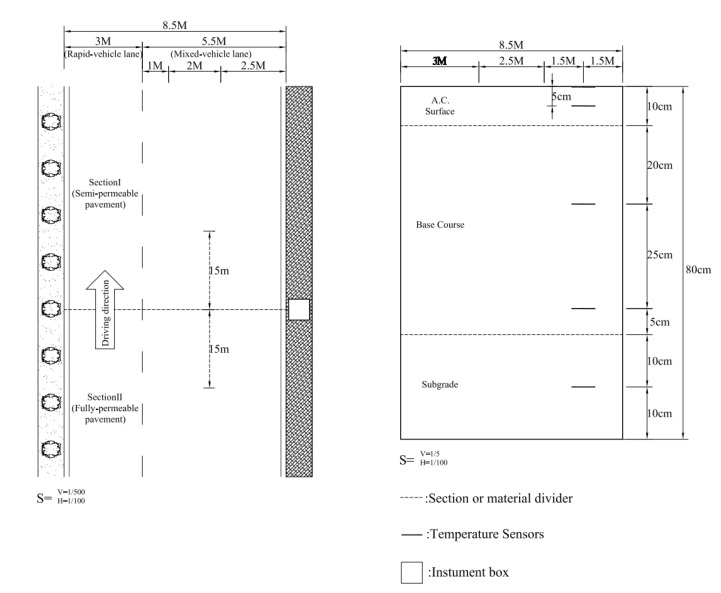
Sketch for the sensor locations in the study.

**Figure 4 materials-14-03431-f004:**
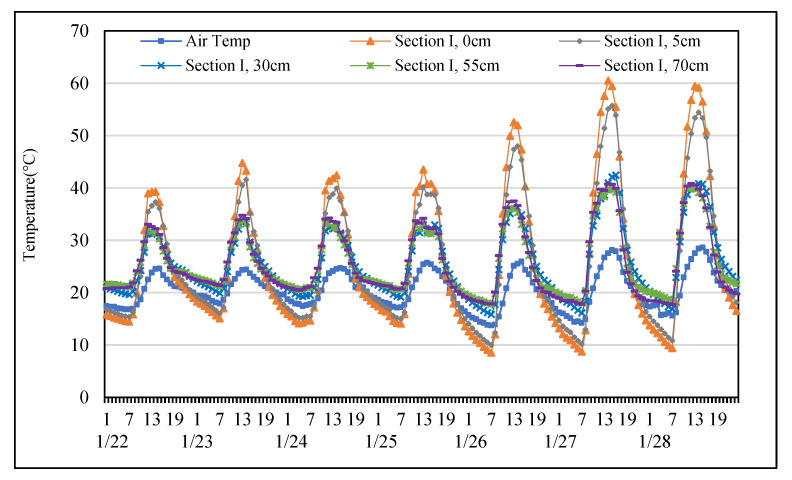
Temperature distribution at each depth of Section I from the 22nd to the 28th of January.

**Figure 5 materials-14-03431-f005:**
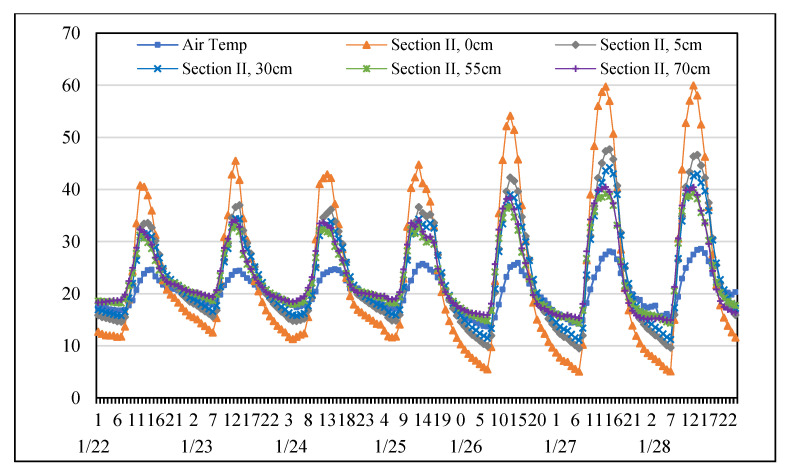
Temperature distribution at each depth of Section II from the 22nd to the 28th of January.

**Figure 6 materials-14-03431-f006:**
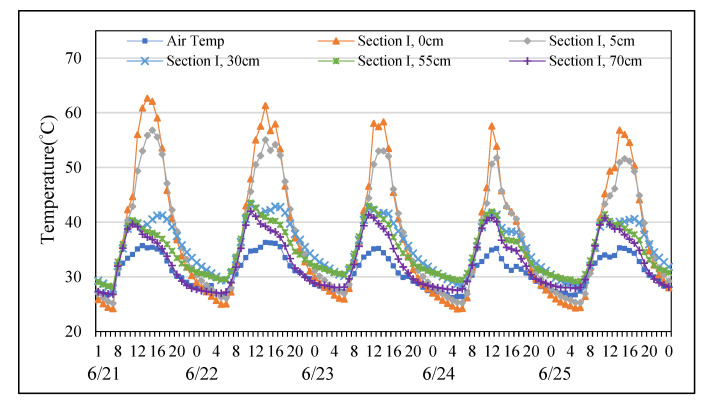
Temperature distribution at each depth of Section I from the 21st to the 25th of June.

**Figure 7 materials-14-03431-f007:**
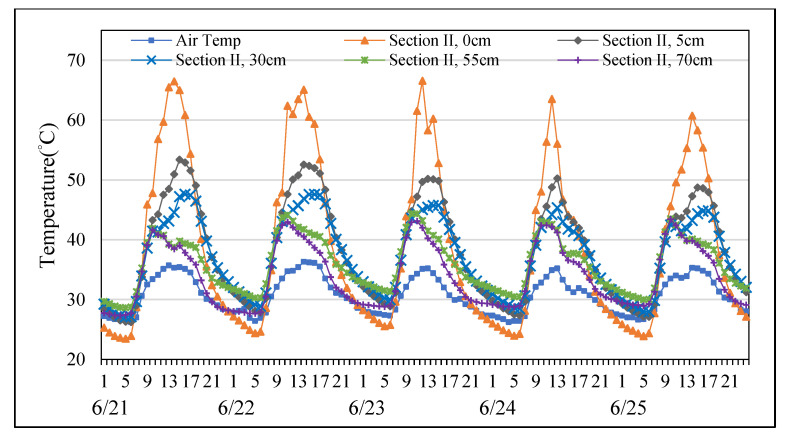
Temperature distribution at each depth of Section II from the 21st to the 25th of June.

**Figure 8 materials-14-03431-f008:**
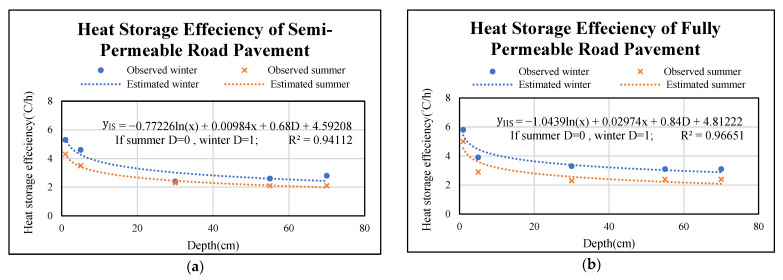
Relationships between heat storage efficiency verse permeable road pavements depth (**a**) Model for semi-permeable road pavement. (**b**) Model for fully permeable road pavement.

**Figure 9 materials-14-03431-f009:**
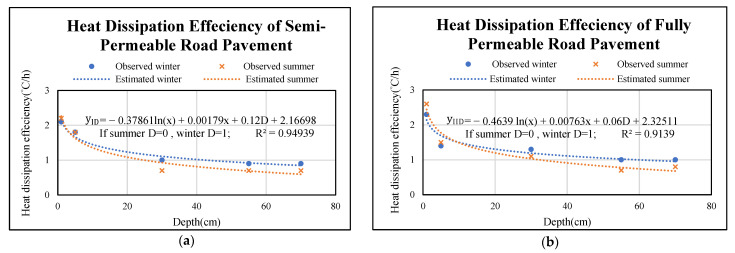
Relationships between heat dissipation efficiency verse permeable road pavements depth (**a**) Model for semi-permeable road pavement. (**b**) Model for fully permeable road pavement.

**Table 1 materials-14-03431-t001:** Material Physical property of semi-permeable and fully permeable pavements.

Course	Physical Property of Materials	Section I	Section II
Surface Course		Porous Asphalt Concrete (PAC)	
	Asphalt Contents (%)	4.8	
	Void ratio (%)	19.5	
	Thickness (cm)	10	
Base Course		Permeable Concrete (PC)	
	Water–Cement Ratio (%)	0.54	
	Amount of Cement(kg)	286	
	Aggregate Volume (m^3^)	0.712	
	Amount of Aggregate (kg)	1920	
	Material Quality (kg)	2335	
	Porosity (%)	24.1	
	Thickness (cm)	50	
Subgrade Course		Filter Layer	
		No. 4 Sieve; Single particle size	
	Thickness (cm)	20	
Type of Geosynthetic	Material	Impermeable Cloth	Geotextile
	Thickness (mm)	1.5	0.74

**Table 2 materials-14-03431-t002:** Material specification of H.D.P.E impermeable cloth in Section I.

Physical Property	Test Method	Specification
Thickness (mm)	ASTM D5199	2.0 ± 10%
Density (g/cm^3^)	ASTM D1505/792	≥0.90

**Table 3 materials-14-03431-t003:** Material specification of geotextile in Section II.

Physical Property	Test Method	Specification
Positive permeable rate (1/S)	ASTM-D4491 or CNS 13298	≥0.6
Apparent opening size (mm)	ASTM-D4751 or CNS 14262	≤0.4

**Table 4 materials-14-03431-t004:** Heat storage and dissipation efficiencies in 22nd–28th January.

	Course		Surface Course		Base Course		Subgrade
Section	Item	Air Temp.	0 cm	5 cm	30 cm	55 cm	70 cm
I	Time interval (h)	−	7	7	7	6	6
	Max. Temp. at rising limb (°C)	26.0	48.9	45.3	35.7	35.5	36.5
	Min. Temp. at rising limb (°C)	16.1	12.0	13.2	18.1	19.7	19.5
	Heat storage efficiency (°C/h)	−	5.3	4.6	2.4	2.6	2.8
	Time interval (h)	−	17	17	17	18	18
	Max. Temp. at dropping limb (°C)	26.0	47.2	43.8	35.0	34.7	35.8
	Min. Temp. at dropping limb (°C)	16.1	11.6	12.8	18.0	19.4	19.3
	Heat dissipation efficiency (°C/h)	−	2.1	1.8	1.0	0.9	0.9
II	Time interval (h)	−	7	7	7	6	6
	Max. Temp. at rising limb (°C)	26.0	49.7	40.0	37.2	34.9	36.2
	Min. Temp. at rising limb (°C)	16.1	9.0	12.7	14.0	16.6	17.4
	Heat storage efficiency (°C/h)	−	5.8	3.9	3.3	3.1	3.1
	Time interval (h)	−	17	17	17	18	18
	Max. Temp. at dropping limb (°C)	26.0	48.0	38.9	36.2	34.2	35.5
	Min. Temp. at dropping limb (°C)	16.1	8.5	12.4	13.7	16.4	17.2
	Heat dissipation efficiency (°C/h)	−	2.3	1.4	1.3	1.0	1.0

**Table 5 materials-14-03431-t005:** Heat storage and dissipation efficiencies in 21th–25th June.

	Course		Surface Course		Base Course		Subgrade
Section	Item	Air Temp.	0 cm	5 cm	30 cm	55 cm	70 cm
I	Time interval (h)	−	8	8	7	6	6
	Max. Temp. at rising limb (°C)	35.5	59.3	53.6	41.7	42.1	40.8
	Min. Temp. at rising limb (°C)	26.5	24.7	25.8	28.8	29.4	27.5
	Heat storage efficiency (°C/h)	−	4.3	3.5	2.3	2.1	2.1
	Time interval (h)	−	16	16	17	18	18
	Max. Temp. at dropping limb (°C)	35.5	60.0	54.2	41.7	42.1	41.0
	Min. Temp. at dropping limb (°C)	26.5	24.9	25.9	29.1	29.4	27.7
	Heat dissipation efficiency (°C/h)	−	2.2	1.8	0.7	0.7	0.7
II	Time interval (h)	−	8	8	7	6	6
	Max. Temp. at rising limb (°C)	35.5	64.5	51.0	44.6	43.4	42.7
	Min. Temp. at rising limb (°C)	26.5	24.2	27.5	28.6	30.2	28.3
	Heat storage efficiency (°C/h)	−	5.0	2.9	2.3	2.4	2.4
	Time interval (h)	−	16	16	17	18	18
	Max. Temp. at dropping limb (°C)	35.5	65.4	51.6	46.4	43.4	42.5
	Min. Temp. at dropping limb (°C)	26.5	24.4	27.8	28.8	30.6	28.6
	Heat dissipation efficiency (°C/h)	−	2.6	1.5	1.1	0.7	0.8

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
