# Peer review of "A Study on Heat Storage and Dissipation Efficiency at Permeable Road Pavements"

_materials, 2021, doi:10.3390/ma14123431_

Round 1
Reviewer 1 Report
- The abstract needs to be improved. You should start with some general information/introduction about your topic (heat storage). Then following with your objective and finally including some major conclusions. Details are not necessary for the abstract. It should be easy to read and give an idea about the content of your paper.
- Line 25-37: You can use some information from this paragraph to start your abstract, but please don`t just copy-paste.
- The introduction looks good. I think you have enough information.
- Figure 1: what do you mean by No.3 of the National Highway?
- Figure 2: What do you mean by Drainage Modified Asphalt Concrete?
- Table 1: What is the "void ratio"?
- Lines 181-185: you should move it to "methods".
- Figures 4-7: is it possible to use thinner lines for these graphics?
- It would be better to use bullets for your conclusions. Right now, it is more like a discussion.
Reviewer 2 Report
General Comments
This paper presented the results of the field test of two types of permeable pavement at the KE DA Road in Pingtung City, Taiwan.
This study has not well organized entirely. Therefore, I have some comments to enhance the paper for readers. Therefore, the following comments should be revised before publishing in the ‘Materials’.
Technical Comments
- The abstract is not well organized, so I recommend that follow that pattern [background] -> [objective] -> [process] -> [results] -> [contribution]
- In ‘Introduction’, the author conducted the literature review to prove the excellence of permeable pavements, However, the author should explain the superiority of research comparing with other researches too.
- In ‘Materials and Methods’, Figures 2 and 3, the width and length scale is not appropriate and should be revised.
- In ‘Results and Discussion’, Figures 4-7, the temperature distribution of each measured data could not be only indicated by one value. The error bar of temperature distribution should be added.
- In ‘Results and Discussion’, The author presented the regression model according to the seasons (Winter and Summer) to calculate the heat storage efficient. I recommend the author develop the multi regression model using temperature and permeable road pavements. Only if it is that way, the heat storage efficient can be calculated in all seasons.
- In ‘Conclusion’ I cannot find the research finding. I think the research finding such as contribution, limitation, and future work should be explained each paragraph in detail.
- In ‘References’ refer to the author guideline, the author should revise style of all references.
Reviewer 3 Report
In general, this paper presents a valid field research dealing with internal heat storage and heat dissipation efficiency of fully-permeable and semi-permeable pavements.
I suggest acceptance of the paper after minor revision.
Specific comments and questions:
- The paper should be proofread, as there are places with grammatical and editorial errors.
- Elaborate why these two types of pavements were selected.
- How often HDPE geomembrane used in pavement construction and in what situations?
- How were depths, at which the measurements were performed, selected?
- Table 1 instead of Ground (first column, last line) it would be more appropriate to use Type of geosynthetic (geotextile and HDPE are not ground).
- Table 2 what does Standard refers to? Table 3 what does Specification and Standard refers to? It is recommended to use the same term for test method, specifications, and standards in both tables.
- Why only physical properties are listed in the tables 2 and 3 as in the text you refer to geotechnical material specifications (row 125 to 130)?
- Change the title of Figure 2, studied pavement types would be more appropriate
- For better flow of the paper in the Results and Discussion section it is recommended to rearrange the order of the subsections. Suggestion: 3.1. Temperature Distributions of Permeable Road Pavements in January; 3.2. Heat storage and dissipation efficiency in January; 3.3. Modelling Permeable Road Pavement Temperatures with Respect to Depths in January; 3.4. Temperature Distributions of Permeable Road Pavements in June; 3.5. Heat storage and dissipation efficiency in June; 3.6. Modelling Permeable Road Pavement Temperatures with Respect to Depths in June;
- Explain the meaning of the index WH in equations (2) and (3), WD in equations (4) and (5); SH equations (6) and (7), SD equations (8) and (9).
- Do you think that these models could be extrapolated to other temperatures and other pavement types?
Round 2
Reviewer 1 Report
Thank you for addressing my comments.
Reviewer 2 Report
The author revised the manuscript reflecting reviewer's comments.